# Effects of Strength Training on Physical Fitness of Olympic Combat Sports Athletes: A Systematic Review

**DOI:** 10.3390/ijerph20043516

**Published:** 2023-02-16

**Authors:** Izham Cid-Calfucura, Tomás Herrera-Valenzuela, Emerson Franchini, Coral Falco, Jorge Alvial-Moscoso, Carolina Pardo-Tamayo, Carolina Zapata-Huenullán, Alex Ojeda-Aravena, Pablo Valdés-Badilla

**Affiliations:** 1Escuela de Ciencias del Deporte y Actividad Física, Facultad de Salud, Universidad Santo Tomás, UST, Santiago 8370003, Chile; 2Sciences of Physical Activity, Sports and Health School, Faculty of Medical Sciences, Universidad de Santiago de Chile (USACH), Santiago 9170022, Chile; 3Martial Arts and Combat Sports Research Group, Sport Department, School of Physical Education and Sport, University of São Paulo, Sao Paulo 05594-110, Brazil; 4Department of Sport, Food and Natural Sciences, Western Norway University of Applied Sciences, 5020 Bergen, Norway; 5IRyS Group, Physical Education School, Pontificia Universidad Católica de Valparaíso, Valparaíso 2581967, Chile; 6Department of Physical Activity Sciences, Faculty of Education Sciences, Universidad Católica del Maule, Talca 3530000, Chile; 7Sports Coach Career, School of Education, Universidad Viña del Mar, Viña del Mar 2520000, Chile

**Keywords:** martial arts, combat sports, sports, resistance training, strength, muscle power

## Abstract

This review aimed to identify the effects of strength training programs on the physical fitness of Olympic combat sports (OCS) athletes. The systematic review included peer-reviewed articles that incorporated interventions that included pre- and post-intervention physical fitness assessment. The search was performed in the SCOPUS, PubMed, and Web of Science databases between April and September 2022. PRISMA and the TESTEX checklist were used to select and assess the methodological quality of the studies. Twenty studies with 504 participants (428 males and 76 females) were included. Significant improvements were found in athletes’ maximal dynamic and isometric strength, muscle power, flexibility, and balance. In addition, improvements in favor of the training groups in specific actions of judo, karate, fencing, and boxing were observed. In conclusion, interventions aimed at the development of muscle strength in OCS, specifically in judo, boxing, karate, wrestling, and fencing, proved to be beneficial at a physical fitness level, resulting in significant increases in favor of the training groups with OCS, which could be used by trainers and coaches to improve the physical performance of athletes.

## 1. Introduction

Currently, there are multiple forms of combat sports; however, only some are part of the Olympic program, as follows: boxing, fencing, judo, karate, taekwondo, and wrestling [1]. Olympic combat sports (OCS) represent between 20% and 25% of all the Olympic medals disputed [2]; with the inclusion of karate in Tokyo 2020, this figure was increased to 26.3% in the last edition of the Olympic Games. Therefore, these modalities may have a relevant impact on each country’s position in the Olympic medal table.

Accordingly, to obtain the most outstanding sports results, OCS athletes seek to achieve their best performance in each Olympic cycle. The OCS are sports with complex physical characteristics, with many physical fitness variables being relevant for sports performance, such as aerobic and anaerobic power and capacity, maximal strength, strength-endurance, muscle power, and flexibility [3,4,5,6,7,8,9]. One of the most relevant qualities being muscle strength and its manifestations (e.g., maximal strength, muscle power, and strength-endurance). In this sense, various studies have reported differences in strength and power between athletes of different competitive levels, in favor of medalists compared to non-medalists [10,11,12,13,14] and between elite athletes versus non-elite ones [15,16,17].

There are the following three major divisions within the OCS based on the main action executed during the matches: striking, weapon-based, and grappling combat sports. In the case of striking and weapon-based OCS, given their particularities (taekwondo, boxing, fencing, and karate), they depend more on muscle power and speed due to the explosive nature of the scoring actions applied with light loads (i.e., own body mass segment or light weapon) in short periods, with longer rest intervals [5,18,19]. Conversely, the grappling OCS (wrestling and judo) possess multiple techniques for displacing and taking down the opponent, which depend mainly on maximal strength, endurance, and power with high force production, as most of the scoring actions are executed with higher loads (i.e., the opponent resistance and his/her body mass). Additionally, these powerful actions being executed are frequently preceded by actions that depend on strength-endurance, such as grips and retentions [3,20,21]. Thus, muscle power and strength-endurance are relevant in both modalities of OCS, as each attack and counter-attack action is executed explosively and repetitively [22,23].

As performance in OCS depends on muscle strength and its manifestations, there is a need to know the effects of muscle strength training on variables relevant to performance in these sports disciplines included in the Olympic program (boxing, fencing, judo, karate, taekwondo, and wrestling). To our knowledge, this is the first systematic review that investigates the effects of muscle strength training in OCS. Therefore, the present systematic review aimed to identify the effects of maximal strength (isometric and dynamic), power, and strength-endurance training programs on physical fitness in OCS athletes.

## 2. Materials and Methods

This review followed the guidelines of the preferred reports for protocols of systematic reviews and meta-analyses PRISMA (Preferred Reporting Items for Systematic Reviews and Meta-analyses) [24], which corresponds to a list 17-item checklist intended to facilitate the development and reporting of a robust protocol for systematic reviews or meta-analyses.

### 2.1. Eligibility Criteria

The inclusion criteria for this review were the following: (i) original articles written in English, Spanish, or Portuguese, (ii) published up to September 2022, (iii) that the population of the study was OCS athletes, regardless of sex; (iv) interventions with muscle strength training (e.g., muscular strength, maximum strength, muscular power, and strength-endurance) with a duration equal to or greater than 4 weeks; (v) who had at least one physical fitness assessment, before and after the intervention; (vi) that their study design was pre-experimental, quasi-experimental, or experimental (randomized controlled trial and non-randomized controlled trial) with pre- and post-assessment. The exclusion criteria were as follows: (i) cross-sectional, retrospective, and prospective studies or those in which the intervention did not focus on OCS; (ii) studies in which the participants were outside the range of 18–35 years of age; (iii) studies that did not correspond to original research publications (e.g., letters to the editor, translations, notes, and book reviews); (iv) duplicate articles; (v) case studies (i.e., studies using only one athlete).

### 2.2. Information and Database Search Process 

The search process was performed between April and September 2022 using the following three databases: PubMed, Web of Science, and Scopus. The medical subject headings (MeSH) from the United States of America National Library of Medicine used bias-free language terms related to strength and OCS. The search string used was the following: (“strength” OR “resistance training” OR “muscular strength” OR “muscle power” OR “explosive strength”) AND (“Olympic combat sports” OR “boxing” OR “fencing” OR “judo” OR “karate” OR “taekwondo” OR “wrestling”).

### 2.3. Selection of Studies and Data Collection

The studies were exported to the Mendeley reference manager (version 1.19.8), where they were filtered once more by selecting the title, abstract, and keywords. Only in some cases, it was necessary to go to the full text of the article. Two authors (ICC, JAM) performed the process independently. Possible discrepancies between the two reviewers on study conditions were resolved by consensus with a third author (THV), it should be noted that this procedure was not necessary during the review. Subsequently, potentially eligible studies were reviewed in full text, and the reasons for exclusion of those studies that did not meet the selection criteria were reported. Study data were extracted by two authors independently, using a form created in Microsoft Excel (Microsoft Corporation, Redmond, WA, USA).

### 2.4. Assessment of Methodological Quality

This phase aimed to detect the risk of bias in each of the selected studies. For this purpose, the tool for the assessment Study quality and reporting in exercise (TESTEX) scale was applied [25]. This instrument is specifically designed for studies with interventions based on physical exercise. The main difference in TESTEX is that there are accommodations for assessment of whether relative exercise intensity remained constant and thus potentially prevented detraining when participants initially adapt to new exercise programs. Assessment of whether evidence-based periodic adjustment of exercise intensity is informed by exercise volume and exercise expenditure. Information on all exercise characteristics (intensity, duration, frequency, and mode) is provided to calculate exercise volume. This tool is a 15-point scale (5 points for study quality and 10 points for reporting) and addresses quality assessment criteria not listed above specific to exercise training studies. This process was carried out by two authors (ICC, JAM) independently, with a third author (THV) acting as arbitrator in doubtful cases, to later be consolidated by two authors (ICC, JAM). This was only necessary in the case of three items 

### 2.5. Data Synthesis

The following data from the selected studies were obtained and analyzed: (i) author and year of publication; (ii) country of origin; (iii) modality: OCS practiced; (iv) sample: total number of participants, mean age, intervention groups, and sex; (v) activities developed in the intervention; (vi) training volume (total duration, weekly frequency, and time per session); (vii) intensity of the intervention; (viii) variables analyzed; (ix) data collection instruments; (x) main outcomes.

## 3. Results

### 3.1. Selection of Studies

The search process is detailed in Figure 1. A total of 3359 records were found in the study identification phase (PubMed = 680, Web of Science = 1198, and SCOPUS = 1481). In the screening phase, duplicates were eliminated, and the studies were filtered by selecting the title, abstract, and keywords, resulting in 1967 references; subsequently, 1905 articles were eliminated for failing to meet two or more inclusion criteria. A total of 61 studies were analyzed in full text, with 19 excluded for having participants under 18 years of age, 11 studies were excluded for not being muscle strength interventions, 4 excluded for having a duration of fewer than 4 weeks, and 1 excluded for not being able to access the full text, 3 excluded for not being an OCS intervention, 3 excluded for not mentioning the age of the participants, and 1 excluded for not detailing the training program. After this process, the number of studies that met all of the selection criteria was 20.

### 3.2. Methodological Quality

The selected studies were analyzed with the TESTEX scale. The total number of studies obtained 45% of the total scale score (15 points), which can be seen in Table 1. Two studies obtained a score of 10/15 [26,27], two obtained 9/15 [28,29], two obtained 8/15 [30,31], five obtained 7/15 [32,33,34,35,36], four obtained 6/15 [37,38,39,40], two obtained 5/15 [41,42], and three obtained 4/15 [43,44,45]. 

### 3.3. Characteristics of the Studies

The studies were carried out on the continent of South America, Europe, Asia, and Africa, specifically six in Brazil [28,29,31,32,33,36], two in Lithuania [34,45], two in Spain [26,27], one in Serbia [38], one in Denmark [44], one in France [37], one in India [41], one in Turkey [30], one in Iran [39], one in Korea [40], one in China [35], and one in Egypt [43], respectively. Regarding the OCS practiced, seven studies were interventions with judo [28,29,31,33,36,37,38], five studies with karate [27,30,39,43,44], five with boxing [32,34,35,40,45], two with wrestling [41,42], and one study with fencing [26]. No articles were found that met the inclusion criteria for the taekwondo modality. Table 2 shows the summary of characteristics, effects, and variables analyzed in each selected study.

### 3.4. Sample Characteristics

Thirteen studies had 8–21 participants[26,31,32,33,34,35,36,37,38,39,40,43,45], three studies with 21–23 participants [27,42,44], three studies with 39–40 participants [28,29,41], and one study with 120 participants [30]. This totaled a sample of 504 athletes, 428 male, and 76 female, with a mean age of 21.5 years.

Another characteristic reported by the studies is related to the competitive level of the sample. Five studies indicated their participants were experienced judo athletes competing nationally [28,29,31,33,37]. One study indicated that its participants were elite-level judo athletes [38]. One study indicated that its participants were national and international level judo athletes [36]. Five studies reported that their participants were karate athletes who competed at the national level [27,30,39] or at the university level [43,44]. Two studies indicated that their participants were amateur boxers belonging to the Olympic team from Brazil [32] or Korea [40]. One study indicated that their athletes were boxers of international and national levels [35]. One study indicated that its participants were amateur boxers who competed locally [34]. One study reported that its participants were elite boxers from the Lithuanian national team [45]. Two studies indicated that their participants were elite-level wrestlers [42] and wrestlers with experience of at least five years [41]. One study indicated that its participants were national-level youth fencers from Spain [26].

### 3.5. Interventions Performed and Dosage

Regarding the intervention groups, five studies had two analysis groups, one experimental group that participated in a strength training intervention and one control group [26,35,37,41,42]. Three studies had three groups, two experimental, and one control [28,29,30]. Twelve studies were of quasi-experimental design, where seven studies only had one training group [32,33,34,36,38,40,43,45], two studies had two training groups [27,31], and only one study had three training groups [44].

In three studies, the activities of the control group were not reported. However, they mentioned that they did not participate in the training program [29,37,42]. The rest of the studies practiced specific fencing movements, tactical exercises in pairs at competition intensity, simulated competitions, individual training, and flexibility exercises targeting the lower limbs [26]. Other control groups performed a technical training program in circuits with various fighting techniques [41], performed traditional strength training [35], and continued their specific judo training [28] and habitual karate training [30].

Regarding the activities performed in strength training programs in OCS, one study executed specific strength training [37], one study prescribed a combined resistance training, throw exercises, jump exercises, and judo-specific techniques [36], and three studies developed a combined resistance training and judo-specific techniques [31,38,42]. Two studies prescribed resistance training and specific striking techniques [40,44], five studies included resistance training [28,29,32,41,42], one study prescribed traditional strength and speed strength training [35], three studies developed combined resistance training and plyometric training [26,39,45], two studies used plyometric training [27,34], one study executed a plyometrics and Pilates training [30] and one study prescribed an exercise program using a Swiss ball [43].

Regarding the training supervision, three studies had specialty professionals in charge of the sessions [26,32,37], whereas seventeen studies did not report supervision of the training [27,28,29,30,31,33,34,35,36,38,39,40,41,42,43,44,45]. The duration of the interventions was diverse, ranging from 4 to 8 weeks [27,30,31,32,34,38,41,42,43,45], from 9 to 13 weeks [26,28,29,35,36,37,39], and from 16 to 20 weeks [33,40,44]. The training frequency for the interventions ranged from 2 to 6 weekly sessions. Three studies reported the duration of their sessions for technical training being from 45 to 165 min for the control group [26,28,41].

Only four studies detailed the duration of their sessions for the experimental group protocol, being 20 min [42], 90 min [40], 120 min [41], and 90–120 min [36]. Twelve studies did not report the duration of their training sessions [27,29,31,32,33,34,35,37,38,39,44].

Franchini et al. [33], Redondo et al. [26], Saraiva et al., [29], and Stojanovic et al. [38] used intensities of 70–90% of one repetition maximum (1RM). Kaya [41] used loads of 80–100% of their maximum capacity in the exercises. Čepulenas et al. [45] based the intensity of the intervention on three stages, general strength with 80–90% of 1RM, strength-speed with 60–80% (strength), and 20–40% (speed) and speed strength with 20–40% (strength) and 60–80% (speed). On the other hand, Franchini et al. [31] implemented intensities based on the 1RM with ranges of 3–5 and 15–20 repetitions. Similarly, Saraiva et al. [28] and Voigt & Klausen [44] used ranges of 6–10 repetitions of the 1RM. Aminaei et al. [39] used intensities of 20–80% of 1RM in both training groups, with jump heights of between 30 and 50 cm for the plyometric group. In turn, Soñen et al. [27] used 10–60 cm height intensities for the unipodal and bipodal group jumps, respectively.

Continuing with the studies’ training intensity, Bruzas et al. [34] implemented exercises with 15% body weight and external weights of 1–1.5 kg. Bu [35] used exercises with intensities from 5 to 25 kg and their own body weight. Kim et al. [40] based their intervention on exercises with 50–70% of 1RM, elastic bands of different resistances, and medicine balls of 3–5 kg. Loturco et al. [32] used the optimal power load in their athletes’ workouts. Finally, five studies did not report the intensity of their workouts [30,36,37,42,43].

### 3.6. Variables Analyzed and Data Collection Instruments

The selected studies used different indicators to evaluate the effects of the interventions; regarding specific performance assessments, one study measured the load mobilized by two throwing techniques (Morote Seoï Nage and Osoto-Gari) with a judo-specific machine together with the scoring of technical quality through a committee of experts watching a recording made with a 25 Hz video camera [37]. Similarly, Nagla et al. [43] performed an analysis of Gankaku kata through five accredited judges of the Egyptian Karate Federation. Saraiva et al. [29], Franchini et al. [31], and Marques et al. [36] evaluated judo performance with the Special Judo Fitness Test. Two studies assessed punch strength in boxers through a diagnostic kit (Kiktest-100) [34,45]. At the same time, Voigt & Klausen [44], Kim et al. [40], and Bu [35] measured punch velocity and power in boxing utilizing a stroboscope (107–104 Hz), three axial acceleration sensors (model 4630, Measurements Specialties, Hampton, VA, USA) and Sony HDR-CX630E 4K digital camera with resistive pressure-sensitive sensor, respectively. One study measured the reaction and movement time in fencing lunge execution using a laser photocell (DSD, Inc., León, Spain) [26].

Five studies assessed maximal strength by 1RM in the horizontal bench press, barbell squat, and seated calf extension exercises [26], bench press, back squat and rowing [31,33,38], bench press and squat [42], only rowing [36], and finally, incline bench press and 45-degree crunches [44]. In addition, Saraiva et al. [29] assessed muscle strength by 10RM test in the bench press, lateral pulldown, military press, push-ups, squats, leg press, leg extensions, and leg curls. On the other hand, Kim et al. [40] assessed maximal strength in the bench press and squat through a multifunction dynamometer (ACE-2000, Ariel Dynamics Inc., Coto de Caza, CA, USA).

Four studies assessed isometric strength via dynamometers [31,33,41,42]. Kaya [41] used a manual digital dynamometer (TKK 5401, Takei Scientific Instruments, Akiba-ku, Niigata, Japan), Franchini et al. [31,33] used a manual hydraulic dynamometer (Jamar, Performance Health, Warrenville, IL, USA), and Özbay et al. [42] used a manual dynamometer (Takei A5001, Takei Scientific Instruments, Akiba-ku, Niigata, Japan) and a leg dynamometer (Takei A5002, Takei Scientific Instruments, Akiba-ku, Niigata, Japan). One study did not report the instrument used for isometric strength assessment [44]. On the other hand, three studies measured trunk strength using an isokinetic dynamometer (Humac Norm, Stoughton, MA, USA) [40], McGill protocol [30], and a maximal number of sit-ups [43].

Regarding explosive strength assessments, two studies used a contact platform (SportJUMP System, DSD, Inc., León, Spain) [26] and Ergojump Pro 2.0 (CEFISE, São Paulo, Brazil) [36]. Soñen et al. [27] and Kaya [41] used an infrared system (Optojump, Microgate, Bolzano, Italy) and a force platform (New Test Power Time, Finland), respectively. Franchini et al. [31] used the Standing Long Jump test. Two studies assessed muscle power through a linear position transducer (T-Force; Ergotech, Murcia, Spain) [32] and a video camera (Raptor-H Digital, Motion Analysis, Westwind, CA, USA) with software (Cortex Version 2.5.0.1160–64 Bit) [39]. One study assessed speed strength through the sitting flat and pushing a solid ball and a 15-s fast pushup test [35].

Three studies measured strength-endurance by using the dynamic judogi chin-up test and isometric judogi chin-up test [31,33], pull-up and push-up tests [42], and a maximum number of repetitions at 70% 1RM in the bench press and rowing exercises [33].

Two studies assessed static-dynamic balance using the mStar balance test [30] and the Stork stand and modified bass tests [43]. Two studies measured the flexibility of their participants using the grand car test and trunk extension Test [43] and a Lafayette goniometer 7514 (Sammons Preston Rolyan, Bolingbrook, IL, USA) [28].

One study used an athletic fitness index of boxers composed of the 30 m sprint, long jump, vertical jump with use of arms, handgrip strength (left and right), 4 kg medicine ball throw (left and right arm), arm stretch for 15 s, leg raises hanging from a bar, abdominal crunches, and tapping test 5 s and 30 s [45]. One study assessed the basic movement ability level through a deep squat, front and back split-leg squat, straight knee leg raise, shoulder joint flexibility, pushup, and rotation stability scored [35].

Finally, Stojanovic et al. [38] measured anaerobic capacity using the running-based anaerobic sprint test and fatigue index using the multistage fitness test. Kaya [41] measured aerobic capacity using the 20 m shuttle run test and anaerobic power using the Wingate test. 

### 3.7. Main outcomes in Physical Fitness

In judo, a significant difference was found for MSN and OSG techniques in the experimental group, which was able to mobilize a higher weight in the judo-specific apparatus after the training program, while the control group was not tested [37]. In addition, the experimental group obtained a better technical quality score after the training program for OSG and MSN, but no change was observed in the control group.

Franchini et al. [31] found similar strength gains in the linear and undulating periodization groups, with increased handgrip strength with the dominant hand and non-dominant hand, isometric judogi chin-up test, 1RM in the bench press, squat and rowing, total weight lifted at 70% 1RM in the bench press and squat, number of throws in SJFT stage B, C, and SJFT index, after the training programs.

In a single group study with judo athletes, after a periodized training program, Franchini et al. [33] found strength increases in the 1RM of the rowing exercise, judogi chin-up isometric test, judogi chin-up dynamic test and increases in upper body anaerobic power, upper body average power and lower body anaerobic power.

Saraiva et al. [28] found a similar range of motion gains in both training groups in shoulder flexion, shoulder extension, shoulder abduction, shoulder adduction, trunk flexion, trunk extension, hip flexion, and hip extension after the training programs.

In a similar study, Saraiva et al. [29] found more significant improvements in the number of throws during the SJFT and SJFT index for the training group that started by performing the lower and then upper body strength exercises, compared to the group that performed the reverse order, obtaining in the number of throws and SJFT index.

Still, regarding judo athletes, Stojanovic et al. [38] found increases, after the training program, in the 1RM of bench press, squat, seated rowing, peak anaerobic power, mean anaerobic power, and fatigue index in elite-level female athletes.

Regarding studies with karate, Aminaei et al. [39] found similar strength gains in the plyometric and cluster groups, with increased 1RM of squat exercise and CMJ with load after the training programs.

Nagla [43] found increases in the number of sit-ups performed and leg strength, as well as in flexibility; the grand car test was used to assess hip flexibility and trunk extension in static balance and dynamic, as well as scoring higher in Gankaku kata performance.

In addition, Pal et al. [30] obtained increases similar in dynamic balance (mStar Excursion Balance Test) for the right anterior, left anterior, right posteromedial, left posteromedial, right posterolateral, left posterolateral for the plyometric group, as for the Pilates group in the right anterior, left anterior, right posteromedial, left posteromedial, right posterolateral, left posterolateral. 

Continuing with studies in karate, Soñen et al. [27] found improvements in countermovement jump, countermovement jump with the non-dominant leg, horizontal jump, horizontal jump with the non-dominant leg, and horizontal jump with the dominant leg for the bilateral training group, in addition to a reduction in countermovement jump asymmetry and horizontal jump asymmetry. Moreover, the monopodal training group obtained improvements in the countermovement jump, horizontal jump, horizontal jump with the non-dominant leg, and horizontal jump with the dominant leg and a reduction in the asymmetry of the countermovement jump and asymmetry in the horizontal jump. It is important to mention that the use of decimals in the distance of the horizontal jumps may not be precise since only the difference in marks on the ground was considered when performing the jumps.

Finally, Voigt & Klausen [44] showed increases for all three training groups in the abdominal 1RM test with an improvement of 45.8% to 53.1% post-intervention and the incline bench press 1RM of 13.9% to 27.3% post-intervention. In addition, the strength training group combined with light training with a punching bag, and the group that only performed strength training increased abdominal isometric strength. Besides that, the strength training group combined with light training with a punching bag and the group that performed intensive punch training with a punching bag and special punching exercises increased isometric strength in the incline bench press with an elbow angle of 135 degrees. Finally, the strength training group combined with light training with a punching bag increased the maximum linear velocity of the hand during the cross-technique execution and the maximum linear velocity of the shoulder during the cross technique. Similarly, the group that performed intensive punch training with a punching bag and special punching exercises increased maximum linear shoulder velocity during the cross-technique execution, and the group that only performed strength training obtained increases in angular velocity of the elbow during the cross-technique execution.

Concerning studies with boxers, Bruzas et al. [34] found increases in rear-hand low punch strength (*p* < 0.05) without detailing the exact values statistically. In addition, they achieved increases in summative strength and energy production in 3 s and 8 s and summative strength and energy production in the 8x8 s series. On the other hand, Čepulenas et al. [45] obtained increases in rear hand force during the straight punch and low blow and in the front hand in side blow and low blows.

In another study with boxers, Kim et al. [40] found improvements in the exercises of bench press and squat 1RM. They also reported increases in the force of the trunk flexion at 30°/s, trunk extension force at 30°/s, the relative force of trunk strength in flexion at 30°/s, trunk relative strength in extension at 30°/s, in addition to increases in isokinetic power of the right arm in extension at 180°/s, isokinetic power of the left arm in extension at 180°/s, the relative power of the right arm in extension at 180°/s, the relative power of the left arm in extension at 180°/s. Moreover, they also increased strength measured in G units in straight punches and hook punches. Bu [35] observed improvements in the sitting flat pushing solid ball test, strike speed, and impact power. Regarding the values of the impact power, we believe that there was an error in the measurement or analysis of the data since the values expressed in m/s are extremely high to be real. Finally, Loturco et al. [32] reported increases in barbell power in the bench press and jump squat exercises.

Regarding wrestling research, Kaya [41] observed improvements in right maximal isometric handgrip strength and VO_2max_. Özbay et al. [42] did not find significant improvements in any of the variables analyzed in that study.

To conclude, regarding fencing, Redondo et al. [26] found increases in squat jump height and countermovement jump, in addition to finding increases in 1RM in the bench press, 1RM in the squat, and 1RM in the seated calf extension. Moreover, athletes decreased the movement time in the specific lunge action.

### 3.8. Adverse Effects

Another relevant aspect corresponds to the adverse effects of muscular strength interventions in OCS. Five studies reported no injuries to their participants in training judo [28,29,33], karate [27], and fencing [26]. Fourteen studies did not report whether their participants were injured during the interventions, these being in boxing [32,34,35,40,45], karate [30,39,43], judo [31,36,37,38], and wrestling [41,42]. Finally, one study reported injuries to its participants without reporting the reason and exact amount [44]. 

## 4. Discussion

This systematic review aimed to identify the effects of strength training interventions on the physical fitness of OCS athletes. After reviewing 3359 records, twenty articles met the inclusion criteria, and only two studies qualified with 50% or more of the established score for methodological quality. The main result of our review indicates that strength programs applied in OCS athletes, specifically with judo, karate, boxing, wrestling, and fencing athletes, are beneficial regarding physical-fitness outcomes [26,27,28,29,30,31,32,33,34,35,37,38,39,40,41,42,43,44,45].

In judo, Blais & Trilles [37] obtained improvements in the experimental group, which was able to mobilize a higher weight in the judo-specific apparatus after the training program, in addition to obtaining a higher score in technical quality. These findings may be related to the specificity of the training by working strength with a machine for specific judo actions [46,47,48].

Franchini et al. [31] found similar increases in the linear and undulating periodization groups in right-hand and left-hand manual grip strength, isometric judogi chin-up test, 1RM in the bench press, squat and rowing exercises, total weight lifted at 70% of 1RM in the bench press and squat exercises, number of throws in SJFT stage B and C and SJFT index. Therefore, both protocols can improve judo athletes’ strength and anaerobic performance. In this regard, there is a general tendency to prefer undulating periodization because the athlete would not be exposed to high intensities continuously, which may allow him/her to accumulate less neural fatigue and facilitate the process of adaptation to training recovery and super-compensation [49]. However, further research with more extended periods of training is needed to validate this hypothesis.

Similarly, in another study, Franchini et al. [33] observed increases in the 1RM of rowing exercise, judogi chin-up isometric test, judogi chin-up dynamic test, and increases in upper body anaerobic power and lower body anaerobic power after a periodized judo training with a duration of 18 weeks (general period of 7 weeks and a specific period of 11 weeks). These results reflect that the adaptations obtained by the training are specific to the demands of judo since there was a more significant increase for strength-endurance adaptations in the variables analyzed than for maximal strength in the only exercise that involved a pulling action [47].

Saraiva et al. [29] found improvements in the number of throws during the SJFT and SJFT index for the training group that started by performing the lower and upper-body strength exercises compared to the group that performed the reverse order. In another similar study, Saraiva et al. [28] found an increase in the range of motion in shoulder flexion and extension, shoulder abduction and adduction, trunk flexion and extension, and hip flexion and extension for both the training group that started with the upper and then lower body strength exercises and for the training group that performed the reverse order.

The order of the sequence of exercises is a critical factor in the training programming since the exercises performed at the beginning have been shown to obtain a higher performance due to a lower accumulation of fatigue [50]. The increased performance in the SJFT test for the group that started with the lower body strength exercises may be because this test places greater emphasis on the lower limbs as opposed to what occurs during combat, where the muscles of the upper limbs must withstand more significant fatigue [28]. However, the fact that they were not elite athletes may have increased sensitivity to chronic responses after the training program. Regarding the similar gains in range of motion for both the group that started the strength exercises with upper and then lower body and for the group that performed the reverse order, it is essential to mention that they performed the same exercises based on machines and free weights. It has been suggested that when strength exercises are performed through a full range of motion and involve both agonist and antagonist muscle groups, flexibility is improved [51] because strength training is essential, from a neuroanatomical perspective, a form of proprioceptive neuromuscular facilitation (PNF) stretching (where a passive full ROM stretch follows a pre-contraction of a muscle) [52].

Finally, Stojanovic et al. [38] found increases in the 1RM of bench press, squat, and seated rowing exercises, as well as increases in peak anaerobic power, mean anaerobic power, and fatigue index after 8 weeks of a pre-competitive training period in elite female athletes. These results confirm that moderate volume strength training, based on two 3-week periods of gradual load increase and one of gradual load reduction, can generate positive adaptations in performance variables in elite female judo athletes. However, the increases in strength variables are not surprising, considering that the athletes had little experience in resistance training. Therefore, it should be taken into account that the gains could be smaller for judo athletes with a more extended history of resistance training [38].

Regarding karate, five studies obtained improvements in different performance variables [27,30,39,43,44]. Aminaei et al. [39] found increased strength in the 1RM of the squat exercise and power in the loaded CMJ exercise. However, no significant differences were found between the cluster and plyometric training groups. A possible explanation may be based on the fact that both methodologies were able to increase the applied force in the spectrum of the force-velocity curve of the subjects of [53] since cluster training through the seconds of rest between repetitions helps to resynthesize part of the phosphocreatine (PCr) depleted in the muscle cell, reducing fatigue and consequently producing higher values of strength and power compared to traditional training without rest between repetitions [54]. In contrast, plyometric training increases the speed of transition from eccentric to concentric phase activation, recruiting more motor units and producing more force, generating adaptations in the musculoskeletal system, involving the muscle spindles, and affecting the viscoelastic properties of the muscles, and, consequently, generating increases in maximum power [55]. Therefore, both methodologies can improve strength and power levels in karate athletes.

Nagla et al. [43] reported increases in the abdominal and leg strength test, as well as in flexibility in the grand car test and trunk extension test. In addition, athletes improved static and dynamic balance, obtaining a higher score in the Gankaku kata performance. The authors attributed these results to the effect of the Swiss ball exercises that contributed to improving the strength of the abdominal and back muscles (trunk muscles) as the main center of movement distribution during the execution of techniques that led to a higher rate of maintaining trunk stability and subsequently the balance of the whole body [43].

Pal et al. [30] found increases in dynamic balance (mStar Excursion Balance Test) and increases in core strength (McGill Test) for both the plyometric group and the Pilates group, with the plyometric group showing a significant improvement compared to the Pilates group and the control group when comparing the pre and post values. Although the Pilates group resulted in greater spinal stability and neuromuscular coordination, observing an improvement in dynamic balance compared to the control group, the plyometric group obtained more significant increases in core strength and dynamic balance than the Pilates group from the point of view of sports performance. This may be due to improved lower extremity muscle contraction or changes in proprioception and neuromuscular control [56] because it is essential in improving balance.

Soñen et al. [27] found improvements in explosive strength in the countermovement jump, countermovement jump with the non-dominant leg, horizontal jump, horizontal jump with the non-dominant leg, and horizontal jump with the dominant leg for the bilateral training group. In addition, the monopodal training group improved the countermovement jump, horizontal jump, horizontal jump with the non-dominant leg, and horizontal jump with the dominant leg. Both training groups could reduce their countermovement jump asymmetries and horizontal jump asymmetries. Although both training groups increased their performance, the subjects who followed a bipodal training methodology improved more. Based on Soñen et al. [27], both training methodologies improve explosive strength. They could be suitable for modalities in which strength is exerted in the horizontal vector, such as karate, by producing a positive effect on performance. Based on the principle of specificity, if vertical vector strength is predominate in the sports modality or the objective is to improve vertical jumping, the methodology performed bilaterally can be a good resource [47].

Finally, Voigt & Klausen [44] showed increases for all three training groups in the abdominal 1RM test and the incline bench press 1RM. The strength training group combined with light training with a punching bag, and the group that only performed strength training also increased abdominal isometric strength, while the strength training group combined with light training with a punching bag and the group that performed intensive punch training with a punching bag and special punching exercises increased isometric strength in the incline bench press with an elbow angle of 135 degrees. In addition, the strength training group combined with light training with a punching bag increased the maximum linear velocity of the hand and shoulder during the blow; similarly, the group that performed intensive punch training with a punching bag and special punching exercises increased the maximum linear velocity of the shoulder during the blow, and the group that only performed strength training obtained increases in the angular velocity of the elbow during the blow. These findings show that strength training alone does not influence the speed of a non-loaded skilled movement, such as a punching motion. However, the exercises performed are questionable because they lack specificity regarding the sports movement evaluated. Instead, specific punch training may have developed a skill that increases the velocity of an unloaded punching movement through a consecutive pattern of segmented movements, exploiting a stretch-shortening cycle in the shoulder flexors and extensors, when combined with progressive strength training, can become more beneficial [44].

Regarding the studies on boxing, three articles found increases in the force of the rear-hand low blows [34], straight and low blow punches with the main hand [45], side and low blows punches with the front hand [45], and straight and hook [40]. One study found increases in punch speed and power [35]. In addition, Bruzas et al. [34] reported increases in the summative force and energy production in 3 s and 8 s and the summative force and energy production in the 8 × 8 s series. These findings suggest that it is possible to generate increases in the specific performance of boxers through muscular strength training using high and light loads. Plyometric training with external weights only generated an increase in the maximum rear hand low punch power; it is possible that the external weights were too heavy to generate power increases across all punches [34]. Besides this, Čepulenas et al. [45] showed that the content and structure of a 4-week mesocycle, composed of 40% athletic training and 60% special training, affect changes in specific work capacity in boxers [56]. Similarly, boxing-specific training improved upper-body power on straight and hook punches after 16 weeks, combined with expansibility and free-weight exercises in amateur boxers who are more familiar with strength training [40].

Continuing the findings in boxing, two studies showed increases in muscle strength and power in the bench press, squat, and jump squat exercises [32,40]; increases were also observed in trunk strength in flexion and extension at 30°/s and in isokinetic power of the right and left arm in extension at 180°/s [40]. This highlights the importance of a combination of strength and plyometric training to improve performance in elite athletes and the possible inclusion of training through optimal power load to maximize power production [53] since it has been shown that the power produced by combat athletes in the bench press and jump squat exercises is closely related to the impact and acceleration of the punch [19].

In wrestling, only one study positively affected wrestlers’ performance [41], these being on right-hand handgrip strength and VO_2max_. High-intensity circuit strength training applied for 8 weeks positively impacted aerobic strength development. In contrast, Özbay et al. [42] did not find any improvement in the variables analyzed because the training protocol did not have any variation in the load applied during the 30 days [21] and was intended for maintenance and not for the development of muscular strength.

Finally, in fencing, Redondo et al. [26] reported increased muscular strength in 1RM exercises in the horizontal bench press, back squat, and seated calf extension, in addition to achieving an increase in jump height in squat jump, CMJ, and a reduction for movement time in the specific lunge action. The results of this study showed that a 12-week strength training program for experienced fencers comprised of resistance training and plyometrics exercises, in addition to regular fencing training, was effective in increasing upper and lower extremity strength, as well as fencing performance. Fencing coaches need to consider that strength exercises should be combined with technical training to transfer strength gains to the kinematic parameters of the lunge [26,47]. Still, it should be considered that the sample of this intervention is not so large as to claim that the results above could be easily extrapolated. Therefore, more studies with larger samples are needed to confirm these findings.

Regarding the dosage used in muscle strength interventions in OCS, it can be indicated that they lasted between 4 to 20 weeks, with 2–6 weekly sessions of from 20 to 165 min [40,41,42]. However, 16 studies did not report the duration of the training sessions [26,27,28,29,30,31,32,33,34,35,37,38,43,44,45], whereas intensity was only reported in 15 studies [26,27,28,29,31,32,33,34,38,39,40,41,44,45]. Thus, it is necessary that future research focused on strength training interventions in OCS mention the stimulus characteristics of their training (duration, frequency, session time, and intensity) to improve the analysis of the interventions for their subsequent application.

Regarding the adherence achieved by participants in OCS muscle strength interventions, 2 studies reported adherence equal to or higher than 85% [29,31], whereas 15 studies did not report whether all participants adhered to the intervention weeks [26,27,28,30,32,34,35,37,38,39,40,41,42,43,45]. Only the studies by Franchini et al. [31] and Voigt & Klausen [44] mentioned the reasons for not attending training. In the study by Franchini et al. [31], the absences were due to a change in the judo club (n = 1), lack of time (n = 4), and a change in the competitive calendar that resulted in a different judo training protocol than the experimental groups (n = 2) [31], with only 13 of the 20 initial athletes completing the intervention. In the study by Voigt & Klausen [44], 21 subjects out of 46 completed the training, and the reasons for dropping out were due to lack of interest, exam requirements, illness, and an injury during the intervention.

Due to the low score on the TESTEX scale, it is difficult to interpret the impacts of the training. The training programs had a minimum duration of 4 weeks, and the frequency of the interventions ranged from 2–6 weekly sessions, with a period of between 20 min and 120 min for each session. It is also possible to use different intensities depending on the objective of the strength training (for example, maximum strength, strength resistance, and muscle power).

The main strengths of this review were as follows: (i) the inclusion of three generic databases (SCOPUS, PubMed, Web of Science) for information retrieval, increasing precision and reducing possible bias in the results obtained, and (ii) the consideration of three languages (English, Spanish, and Portuguese) for the selection of studies, which broadened the scope of the search. The main limitations of this review include the following: (i) we have not registered the study protocol a priori on a platform (for example, PROSPERO); (ii) the fact that most of the studies achieved a low grade in the TESTEX; (iii) the diversity of instruments and variables observed, as well as the small number of available studies of high methodological quality; (iv) not finding studies of interventions in taekwondo that met the inclusion criteria, which reduces the generalizability of the information and (v) only having one study with female athletes. In this sense, further research is needed to fully detail the interventions or methodologies used and strength training interventions in female athletes to analyze their adaptations and optimize sports training processes in OCS.

## 5. Conclusions

In conclusion, strength training can improve performance in specific tests of judo, fencing, karate, and boxing, and strength training can improve different manifestations of strength, such as maximum strength, strength-endurance, and muscle power as other variable physical conditions. Therefore, interventions aimed at developing muscular strength in OCS, specifically in judo, boxing, karate, wrestling, and fencing, seem to be beneficial for physical fitness, resulting in an increase in favor of the training groups. Nevertheless, our systematic review did not include studies on the effects of muscle strength training in taekwondo since the existing articles did not meet the inclusion criteria. In addition, only one article included a sample of female athletes, so we recommend that future researchers conduct studies containing muscular strength training interventions in females to analyze their effects on sports performance.

## Figures and Tables

**Figure 1 ijerph-20-03516-f001:**
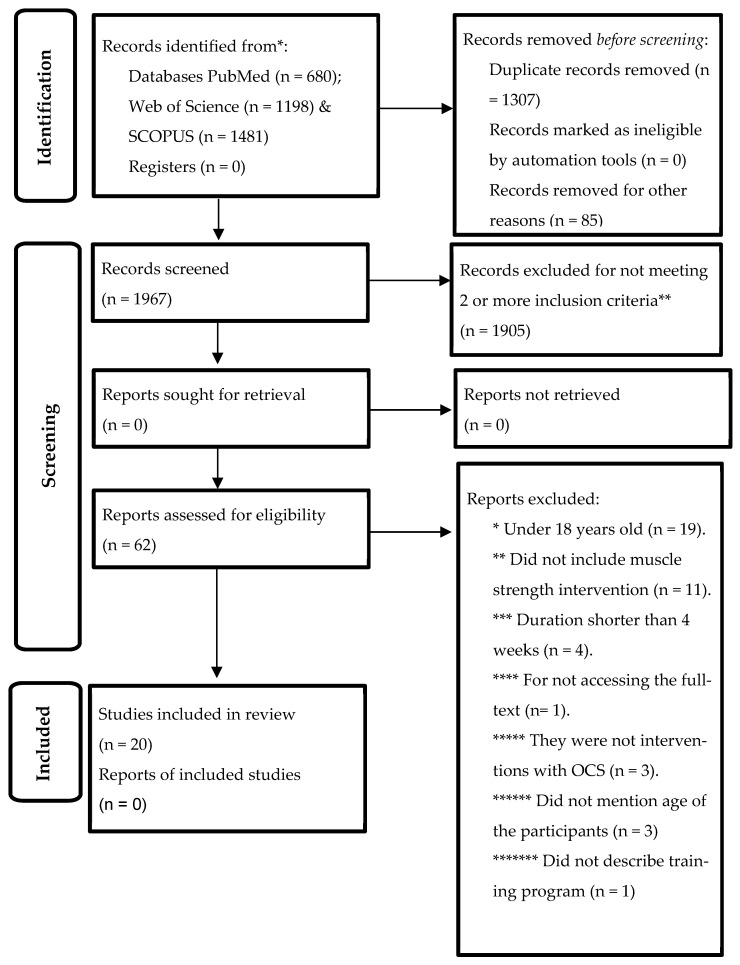
Flowchart of the review process. Based on the PRISMA guidelines [24].

**Table 1 ijerph-20-03516-t001:** Study quality assessment according to TESTEX scale.

Study	Elegibility Criteria Specified	Randomly Allocated Participants	Allocation Concealed	Groups Similar at Baseline	Assessors Blinded	Outcome Measures Assess >85% of Participants *	Intention to Treat Analysis	Reporting of between Group Statistical Comparisons	Point Measures and Measures of Variability Reported **	Activity Monitoring in Control Group	Relative Exercise Intensity Review	Exercise Volume and Energy Expended	Overall TESTEX #
Blais, Trilles (2006) [37]	Yes	Yes	No	No	No	Yes (1)	No	Yes (2)	No	No	No	Yes	6/15
Redondo et al. (2014) [26]	Yes	Yes	No	Yes	No	Yes (1)	No	Yes (2)	Yes	Yes	Yes	Yes	10/15
Pal et al. (2021) [30]	Yes	Yes	Yes	Yes	No	Yes (1)	No	Yes (1)	Yes	No	No	Yes	8/15
Stojanovic et al. (2009) [38]	Yes	No	No	No	No	Yes (1)	No	Yes (2)	No	No	Yes	Yes	6/15
Saraiva et al. (2017) [29]	No	Yes	No	Yes	No	Yes (2)	No	Yes (2)	Yes	No	Yes	Yes	9/15
Soñén et al. (2021) [27]	Yes	Yes	Yes	Yes	No	Yes (1)	No	Yes (2)	Yes	No	Yes	Yes	10/15
Nagla (2011) [43]	No	No	No	No	No	Yes (1)	No	Yes (1)	Yes	No	No	Yes	4/15
Loturco et al. (2018) [32]	Yes	No	No	Yes	No	Yes (1)	No	Yes (2)	Yes	No	No	Yes	7/15
Kaya (2015) [41]	Yes	No	No	Yes	No	Yes (1)	No	Yes (1)	No	No	No	Yes	5/15
Franchini et al. (2015) [31]	Yes	No	No	No	No	Yes (2)	No	Yes (2)	Yes	No	No	Yes	7/15
Bruzas et al. (2018) [34]	Yes	Yes	No	Yes	No	Yes (1)	No	Yes (2)	No	No	No	Yes	7/15
Aminaei et al. (2017) [39]	Yes	No	No	Yes	No	Yes (1)	No	Yes (1)	Yes	No	No	Yes	6/15
Ozbay et al. (2020) [42]	Yes	No	No	No	No	Yes (1)	No	Yes (1)	Yes	No	No	Yes	5/15
Franchini et al. (2015) [33]	Yes	Yes	No	No	No	Yes (1)	No	Yes (2)	No	Yes	Yes	Yes	8/15
Saraiva et al. (2014) [28]	Yes	Yes	No	Yes	No	Yes (2)	No	Yes (1)	Yes	No	Yes	Yes	9/15
Voigt, Klausan (1990) [44]	No	No	No	No	No	Yes (2)	No	Yes (1)	No	No	No	Yes	4/15
Kim et al. (2018) [40]	Yes	No	No	No	No	Yes (1)	No	Yes (2)	No	No	Yes	Yes	6/15
Čepulenas et al. (2011) [45]	No	No	No	No	No	Yes (1)	No	Yes (1)	No	No	Yes	Yes	4/15
Bu (2022) [35]	Yes	No	No	Yes	No	Yes (1)	No	Yes (2)	No	No	Yes	Yes	7/15
Marques et al. (2017) [36]	Yes	No	No	No	No	Yes (1)	No	Yes (2)	Yes	No	Yes	Yes	7/15

* three points possible: one point if adherence >85%, one point if adverse events reported, one point if exercise attendance is reported. ** Two points possible: one point if primary outcome is reported, one point if all other outcomes reported. # total out of 15 points. TESTEX: tool for the assessment of study quality and reporting in exercise.

**Table 2 ijerph-20-03516-t002:** Characteristics and effects of studies about muscle strength training in Olympic combat sports.

Study	Country	OCS Modality	Groups (n)	Mean Age (year)	Activities in the Intervention and Control Groups	Training Volume			Training Intensity	Main Outcomes
						TD (Weeks)	Fr (Weekly)	TPS (Min)		
Blais, Trilles (2006) [37]	France	Judo	EG: 10 CG: 10	22.0 23.0	EG: specific strength training CG: no training	10	2 NR	NR NR	NR NR	EG vs. CG: ↑ kg used in MSN and OSG. EG vs. CG: ↑ Technical score in MSN and OSG.
Redondo et al. (2014) [26]	Spain	Fencing	EG: 6 CG: 6	24.8 22.3	EG: RT and PLY CG: regular technical training	12	2	NR From 45 to 165	70–75% 1 RM NR	EG vs. CG: ↑ SJ and CMJ, ↑ 1RM HBP, 1RM SQ, and 1RM SCE. EG vs. CG: ↑ movement time in the lunge.
Pal et al. (2021) [30]	India	Karate	EG1: 40 EG2: 40 CG: 40	21.0	EG1: PLY EG2: PTE CG: habitual training	8	3 3 NR	NR NR 60	NR NR NR	PLY and PTE vs. CG: ↑ dynamic balance and core strength. PLY vs. PTE: ↑ dynamic balance and core strength.
Stojanovic et al. (2009) [38]	Serbian	Judo	EG: 11	19.9	EG: RT, LCR and TE-TA	8	3	NR	70–80% 1RM	↑ 1RM in BP, SQ and Row. ↑ Peak power ↑ Mean power ↑ Fatigue index
Saraiva et al. (2017) [29]	Brazil	Judo	EG1: 13 EG2: 13 CG: 13	20.7 20.2 20.2	EG1: RT being first BP, LPD, MP, BC and SQ, LP, LE, LC. EG2: RT in reverse order of exercises of EG1 CG: NR	12	3 3 NR	NR NR NR	80–90% 10RM 80–90% 10RM NR	EG1 and EG2 vs. CG: ↑ in number of takedowns in SJFT and SJFT index. EG2 vs. EG1: ↑ in number of takedowns in SJFT and SJFT index.
Soñén et al. (2021) [27]	Spain	Karate	EG1: 11 EG2: 11	21.8	EG1: BI-PLY EG2: U-PLY	6	2	NR	DJ: 10, 20, 30, 40, 50, and 60 cm.	BI-PLY group: ↑ CMJ, ND-CMJ, HJ, ND-HJ, and D-HJ. U-PLY group: ↑ CMJ, HJ, ND-HJ, D-HJ.
Nagla (2011) [43]	Egypt	Karate	EG: 12	19.1	EG: SBT	8	4	NR	NR	↑ AMS, BMS, HMF, SMF, and BSD. ↑ Gankaku kata scoring
Loturco et al. (2018) [32]	Brazil	Boxing	EG: 12	28.1	EG: BP and JS with OPL	7	From 1 to 3	NR	OPL	↑ MP, MPP, and PP in BP ↑ MP, MPP, and PP in JS
Kaya (2015) [41]	Turkey	Wrestling	EG: 20 CG: 20	21.6 21.1	EG: PCC CG: regular technical training	8	3 3	120	80–100% 1RM NR	EG vs. CG: ↑ Right hand grip isometric force EG vs. CG: ↑ VO_2max_
Franchini et al. (2015) [31]	Brazil	Judo	EG1: 7 EG2: 7	From 18 to 35	EG1: LT EG2: UT	8	3	NR	3–5 RM and 15–20 RM	LT: ↑ ISMG, IJCP, 1RM in BP, SQ, and RG, TWL at 70% 1RM in BP and SQ. ↑ Throws in SJFT stage B and C. ↑ SJFT index UT: ↑ ISMG, IJCP, 1RM in BP, SQ, and RG, TWL at 70% 1RM in BP and SQ. ↑ Throws in SJFT stage B and C. ↑ SJFT index
Bruzas et al. (2018) [34]	Lithuania	Boxing	EG: 8	22.3	EG: PLY	4	3	NR	15% BM and external weights of 1–1.5 kg	↑ Rear-hand low punch strength. ↑ Summative force in 3 s and 8 s. ↑ Power production in 3 s and 8 s.
Aminaei et al. (2017) [39]	Irán	Karate	EG1: 9 EG2: 9	18.2	EG1: PLY EG2: CT	9	3	NR NR	20–85% 1RM and 30–50 cm JH 20–85% 1RM	PLY: ↑ SQ 1RM and Loaded CMJ Muscle Power CT: ↑ SQ 1RM and Loaded CMJ Muscle Power
Özbay et al. (2020) [42]	Turkey	Wrestling	EG: 12 CG: 11	20.4 19.4	EG: CFT CG: no training	30 days	6	20 NR	NR	No significant improvements in the variables analyzed
Franchini et al. (2015) [33]	Brazil	Judo	EG: 10	19.9	EG: GP, SP, and CX	18	3	NR	70–90% 1RM	↑ 1RM in Rowing ↑ IJCP ↑ DJCP ↑ UB peak power ↑ LB peak power
Saraiva et al. (2014) [28]	Brazil	Judo	EG1: 13 EG2: 13 CG: 13	20.7 20.2 20.2	EG1: RT being first BP, LP, MP, BC, and SQ, LP, LE, LC. EG2: RT in reverse order of exercises of EG1 CG: regular technical training	12	3 3 8	NR NR 120	10 RM 10 RM NR	EG1: ↑ SF, SE, SAB, SAD, TF, TE, HF y HE EG2: ↑ SF, SE, SAB, SAD, TF, TE, HF y HE
Voigt, Klausan (1990) [44]	Denmark	Karate	EG1: 8 EG2: 8 EG3: 5	18 to 21	RT and PBT PBT and SPE RT and PBT	20 20 16	3 3 3	NR NR NR	6 RM NR 6 RM	EG1: ↑ 1RM in 45-degree AC and IBP ↑ MVC Abdominal ↑ MVC on IBP with elbow angle at 135° ↑ Vh max, and Vs max. EG2: ↑ 1RM in 45-degree AC and IBP ↑ MVC in IBP with elbow angle at 135 ↑ Vs max. EG3: ↑ 1RM in 45-degree AB and IBP ↑ MVC Abdominal ↑ Vh max, and EAV.
Kim et al. (2018) [40]	Korea	Boxing	EG: 15	23.4	PCT, ETT, MBE, and BST	16	3	90	PCT: 50–70% 1RM ETT: elastic tubes (green, blue and black) MBE: 3, 4, and 5 kg	↑ BP and SQ strength ↑ Relative TS in EXT and FLEX at 30°/s ↑ TS (Nm) in EXT and FLEX at 30°/s ↑ IP (w) of right and left arm in EXT at 180°/s ↑ RP (%BW) of right and left arm in EXT at 180°/s ↑ Power of direct punch and hook punch.
Čepulenas et al. (2011) [45]	Lithuania	Boxing	EG: 10	22.5	RT, PLY, and WS	4	NR	NR	20–90% 1RM	↑ BHP in straight blow, side blow and body blow. ↑ FHP in straight blow (Jab), side blow (hook to the body) and body blow (straight to the body).
Bu (2022) [35]	China	Boxing	EG: 10 CG: 10	From 21 to 23	RT and SST Only RT	12	3	NR	RT: 5–25 kg SST: BM – 2.5 kg RT: 5–25 kg	↑ Sitting flat pushing solid ball ↑ Punching speed ↑ Punching power
Marques et al. (2017) [36]	Brazil	Judo	EG: 21	21.8	Block periodization	13	4–5	90–120	NR	↑ SJFT index

OCS: olympic combat sports. TD: total duration. FR: frequency. TPS: time per session. NR: not reported. EG: experimental group. CG: control group. ↑: significant improvements. MSN: morote Seoi Nage. OSG: osoto-gari. 1RM = one-repetition maximum. RT: resistance training. PLY: plyometric. SJ: squat jump. CMJ: countermovement jump. HBP: horizontal bench press. SQ: squat. SCE: seated calf extension. PTE: pilates. LCR: long continuous running. TE-TA: technical-tactical training. BP: bench press. LPD: lat pull-down. MP: military press. BC: biceps curls. LP: leg press. LE: leg extension. LC: leg curl. SJFT: special judo fitness test. BI-PLY: bipodal plyometric. U-PLY: unipodal plyometric. DJ: drop jump. ND-CMJ: non-dominant countermovement jump. HJ: horizontal jump. ND-HJ: non-dominant horizontal jump. D-HJ: dominant horizontal jump. SBT: swiss ball training. AMS: abdominal muscle strength. BMS: back muscle strength. HMF: hip muscle flexibility. SMF: spine muscle flexibility. BSD: balance static-dynamic. JS: jump squat. OPL: optimal power load. MP: medium power. MPP: mean propulsive power. PP: peak power. PCC: physical conditioning circuit. VO_2max:_ maximum oxygen consumption. LT: linear periodization group. UT: undulating periodization group. ISMG: isometric strength manual grip. IJCP: isometric judogi chin-up test. RG: rowing. TWL: total weight lifted. BM: body mass. CT: cluster training. JH: jump height. CFT: crossfit training. GP: general period. SP: special period. CX: complex training. DJCP: dynamic judogi chin-up test. UB: upper body. LB: lower body. SF: shoulder flexion. SE: shoulder extension. SAB: shoulder abduction. SAD: shoulder adduction. TF: trunk flexion. TE: trunk extension. HF: hip flexion. HE: hip extension. PBT: punching bag training. SPE: special punching exercises. AC: abdominal crunches. IBP: incline bench press. MVC: maximum voluntary contraction. Vs max: linear velocity of the shoulder. Vh max: linear speed of the hand. EAV: elbow angular velocity. PCT: power circuit training. ETT: elastic tubing training. MBE: medicine balls exercises. BST: boxing-specific training. TS: trunk strength. EXT: extension. FLEX: flexion. IP: isokinetic power. RP: relative power. WS: weighted strikes. BHP: back hand power. FHP: front hand power. SST: speed strength training.

## Data Availability

All data generated or analyzed during this study will be/are included in the published article as Table(s) and Figure(s). Any other data requirement can be directed to the corresponding author upon reasonable request.

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
