# Peer review of "Effects of Strength Training on Physical Fitness of Olympic Combat Sports Athletes: A Systematic Review"

_ijerph, 2023, doi:10.3390/ijerph20043516_

Round 1
Reviewer 1 Report
Dear Authors
You have written an interesting review paper. The introduction is on point as clearly leads to the main rationale.
The prisma guidelines are strictly followed. The inclusion criteria search keywords and databases used are adequate.
Results are clearly presented with a detailed discussion.
The first reference in the reference list is incomplete - please amend it.
Overall a well-written study with a clear focus and good methodology.
Author Response
Dear Editor,
We would like to thank you for the opportunity to respond to the reviewers. We appreciate their efforts, considering all the comments, and believe their appraisal has strengthened our manuscript. Our replies to their suggestions are provided below (answered point by point).
Sincerely yours,
The authors.

Reviewer 2 Report
Dear authors, first of all, thank you for submitting your article to IJERPH.
The article is a systematic review about the effect of strength training on Physical Fitness of Olympic Combat Sports Athletes. This is a topic of particular importance for training programming and the development of Olympic athletes, and therefore the authors deserve credit for exploring the topic and bringing new insights for coaches of different combat disciplines.
Although the authors have clearly explored the theme, I believe that there are some points that, if improved, can help to improve the quality of the work. Below you can find my comments in detail:
The introduction provides a clear view of the purpose of the article and the same is true of the description of the methodologies used where the authors have done an excellent job.
However, after that there is an endless number of pages with tables. Some of them with little information, others with different types of formatting, which in my opinion greatly affects the reading clarity of the manuscript.
So I suggest that authors might consider reorganizing the information that incorporates the tables making it more appealing / more concentrated in order to facilitate reading and understanding. This is in fact one of the biggest criticisms I can make of the work, which, as I mentioned earlier, is pertinent.
In addition, and taking into account that clearly exploring the results of the review in the discussion, it would be highly pertinent if you could provide a recommendation regarding strength training in the light of the results found. What can the coach do with the information from this study? What recommendations? Are the recommendations the same for all combat modes?
Author Response

(The authors gave the same response as above.)

Reviewer 3 Report
To analyze whether strength training manages to increase strength it is not necessary to do a literature review starting with 1900 articles.
In the introduction, the authors do not write about "physical fitness" or about their forms of evaluation.
Why hasn't your review been inscribed in "Prospero"?
The initial search terms are very broad and many potential items are located. The inclusion of many sports makes it difficult to find and process information. Although all the sports analyzed are combat sports, the authors should have clarified the objective more.
It's unclear how 1,905 items have been excluded.
Figure 1 does not specify what **is meant in the "excluded records" box.
In results, section 3.8 (3.8. Main outcomes in physical fitness) the authors include excessive information repetition of the authors' results. There are many data that make it difficult to interpret the results. These results should be summarized in tables or only indicate what is relevant, not the numerical data.
Table 3 provides information on some instruments.
In the discussion many data are listed that are not discussion, they are information.
The conclusion of the work is obvious. "Interventions aimed at building muscle strength in OCS, specifically in judo, boxing, karate, wrestling and fencing, appear to be beneficial in a physical fitness, resulting in an increase in favor of training groups. To do otherwise would be absurd.
The conclusion section includes aspects that are limitations of their work. These are not conclusions.
The authors do not respond in their conclusions to their objective. I don't see the answer to the analysis of the effects of maximum force... about physical fitness.
“Therefore, the present systematic review aimed to analyze the effects of maximal strength (isometric and dynamic), power, and strength-endurance training programs on physical fitness in OCS athletes”.
Author Response

(The authors gave the same response as above.)

Reviewer 4 Report
I applaud the authors for this work. Conducting a systematic review such as this is daunting. The topic is relevant and applicable to a wide range of audiences.
Overall, I believe the manuscript is well written and warrants further consideration for publication after minor revisions. Below are a few observations intended to assist the authors in developing a slightly more refined product.
L 32: Delete their
L 44: what is a muscle strength manifestation?
L 57: these powerful actions being executed…
Strength-endurance or strength endurance?
L 64: be more specific the these
Abstract states April and September, Methods states September – which one is correct?
L 90: I am not sure this sentence needs to be offered.
L 141: 11 studies were excluded…..otherwise this portion of the sentence has no subject
L 145: the number of studies that met all of the selection criteria was 20
Perhaps breaking the latter portion of the results into sections with headings to articulate the respective OCS would be helpful for the ingestion of information in a more organized manner.
Reference #1 is questionable or at a minimum less than useful to the reader
I do suggest the authors revisit the manuscript once more to sort out minor grammatical errors or word/phrase choices.
Author Response

(The authors gave the same response as above.)

Reviewer 5 Report
Dear Authors,
Congratulation for an interesting article especially for trainers of Olympic Combat Sports modalities. In our opinion your methodology is correct and highlight a lack of studies of quality. With the poor testext scores it is difficult to interpret the training impacts, but in your discussion, you did an excellent job extracting the maximum information available for practise insight. We have only two recommendations: in the end of the introduction you should justify the originality of your study, and in the table 2, please take care of a uniform formatting of the text and do not forget to signal de standard deviation by a ±
We wish you success in this publication.
Author Response

(The authors gave the same response as above.)

Round 2
Reviewer 2 Report
Dear authors, after making the requested changes, I no longer have reservations about the manuscript. Therefore, I recommend its acceptance.
Author Response

(The authors gave the same response as above.)

Reviewer 3 Report
I appreciate and admire the effort of the authors to improve their manuscript. Unfortunately I still think that putting all the sports together for a review does not clarify the situation.
Also that, to conclude: "strength training can improve performance in specific tests of judo, fencing, karate and boxing, and strength training can improve different manifestations of strength, such as maximum strength, strength-endurance and muscle power as other variable physical conditions", it is not necessary to perform a review nor is it of interest to potential readers.
Author Response

(The authors gave the same response as above.)
